# A Low-power wearable acoustic device for accurate invasive arterial pressure monitoring

Maruchi Kim[1], Anran Wang[1], Srdjan Jelacic[2], Andrew Bowdle[2], Shyamnath Gollakota [1✉] &
Kelly Michaelsen [2✉]

### Abstract

**Background** Millions of catheters for invasive arterial pressure monitoring are placed annually in intensive care units, emergency rooms, and operating rooms to guide medical treatment decision-making. Accurate assessment of arterial blood pressure requires an IV pole-attached pressure transducer placed at the same height as a reference point on the patient's body, typically, the heart. Every time a patient moves, or the bed is adjusted, a nurse or physician must adjust the height of the pressure transducer. There are no alarms to indicate a discrepancy between the patient and transducer height, leading to inaccurate blood pressure measurements.

**Methods** We present a low-power wireless wearable tracking device that uses inaudible acoustic signals emitted from a speaker array to automatically compute height changes and correct the mean arterial blood pressure. Performance of this device was tested in 26 patients with arterial lines in place.

**Results** Our system calculates the mean arterial pressure with a bias of 0.19, inter-class correlation coefficients of 0.959 and a median difference of 1.6 mmHg when compared to clinical invasive arterial measurements.

**Conclusions** Given the increased workload demands on nurses and physicians, our proof-of concept technology may improve accuracy of pressure measurements and reduce the task burden for medical staff by automating a task that previously required manual manipulation and close patient surveillance.

### Plain language summary

Arterial catheters are commonly inserted in hospitalized, critically ill patients to measure blood pressure. For these systems to work properly, a device that measures pressure, called a pressure transducer, must be connected to the catheter, and maintained at the same height as a reference point, usually the heart. So, if the patient moves, the transducer must be manually adjusted by a nurse of physician, adding to the workload of busy clinicians. If not adjusted, this will lead to inaccurate blood pressure measurements. We built a low-power wearable tracking device that uses inaudible acoustic signals to track changes in the patient's position. These height differences can be used to calculate accurate blood pressure measurements automatically. This device can decrease clinician workload by removing the need to move the transducer by hand, allowing providers to focus on other tasks.

[1] Paul G. Allen School of Computer Science and Engineering, University of Washington, Seattle, WA, USA. [2] Department of Anesthesiology & Pain Medicine, University of Washington, Seattle, WA, USA. ✉email: gshyam@cs.washington.edu; kellyem@uw.edu

More than eight million catheters are placed for invasive arterial pressure measurement in the United States with 36% of intensive care unit (ICU) patients receiving one[1,2]. Invasive arterial pressure measurements require insertion of a catheter into an artery of a patient (typically radial artery) and connecting it to a pressure transducer via fluid-filled inflexible plastic tubing. For an arterial pressure measurement to be accurate and clinically relevant, the pressure transducer must be maintained at the level of a hydrostatic reference point on the patient's body. In a supine patient, this is most commonly the level of the patient's right atrium of the heart (phlebostatic axis), but alternative heights may be chosen depending on the vascular bed of clinical interest, such as when measuring intracranial pressure[3]. The transducer is frequently placed in a plastic holder clipped to the intravenous fluid (IV) pole next to the patient and moved up and down when the patient's heart height changes.

Maintaining the pressure transducer in the correct position is a task requiring considerable vigilance and manual manipulation by healthcare providers as there is no alarm system to detect discrepancies between the transducer height and patient's hydrostatic reference point. Height discrepancies can lead to inaccurate blood pressure measurements and result in erroneous treatment decisions[4]. In clinical practice, the provider visually estimates the location of the patient's hydrostatic reference point and then attaches the pressure transducer at the same height to the IV pole at the bedside. This manual technique is prone to errors and prior work showed that transducers were placed an average of 11 cm away from the proper location, resulting in inaccurate blood pressure readings[5]. Even when providers used a laser leveling tool, their measurement errors ranged from 3.6 to 5.6 cm above or below the proper level depending on patient position[6]. Further, only 3.4% of clinicians could correctly identify the hydrostatic reference point when surveyed[7]. These errors can lead to substantial over or under estimation of blood pressure, as a 10 cm error in pressure transducer height is equivalent to a 7.4 mmHg error in blood pressure measurement.

If bed height or patient position changes or a transducer falls to the floor, the blood pressure displayed will be erroneous. Automating this currently manual task of adjusting the transducer height would reduce errors and improve patient care. Nursing and anesthesia provider task overload can be a serious problem, suggesting that automation of routine tasks is valuable and may help to prevent errors and adverse outcomes[8–10].

In this paper, we present a height tracking system that uses a small, wireless, low-power wearable device and a speaker array that can automatically track the height of the hydrostatic reference point and make corrections to the pressure measurements without manually adjusting the height of the pressure transducer. The low-power wearable device is equipped with a Bluetooth chip, an accelerometer, and a microphone, and can be affixed to a patient via an electrocardiogram (ECG) electrode (Fig. 1). The speaker array (localizer) is comprised of four speakers that emit inaudible acoustic signals that are captured by the wearable device. Using these signals, the height tracking system continuously and accurately determines the three-dimensional position of the wearable device on the patient, and uses the z-plane (vertical) information to correct the blood pressure measurements. The goal of this study was to ensure that the height tracking system could accurately measure mean arterial pressure within 3 mmHg of the current clinical standard measurement while automatically adjusting for changes in heart height. Following full integration with vital sign monitors, our proof-of-concept system could increase the accuracy of invasive pressure measurements while eliminating an error prone manual task for task overloaded care providers.

## Methods

**Concept and prototype.** Early methods to align the pressure transducer to the appropriate height include qualitative inclinometric devices affixed between the patient and the transducer or a wheeled mechanical device which moved from the patient's hydrostatic reference point to a pressure transducer[11,12]. Optical techniques that shine a light from the height of the pressure transducer onto the patient's hydrostatic reference point or a retractable inclinometer for optimal positioning have been described[13–16] but require the provider to manually readjust the transducer when the patient reference point changes. Optical distance measurement devices placed beneath the bed can be used to measure bed height from the ground. However, the head of the bed can be raised or lowered and the patient can move around in bed, leading to changes in the hydrostatic reference point height that would not be assessed by such devices. Finally, one may imagine a camera-based system that can provide depth information, but camera systems raise considerable privacy issues in the clinical setting.

Our design has two key hardware components: a wireless low-power wearable and a localizer device. The wearable device is a 2.5 cm diameter patch affixed to the patient which can be used to determine its spatial position from the localizer device that is attached at a fixed height to an IV pole (Fig. 1a, b). The localizer device does not interfere with the clinical use of the IV pole since it is attached above the hooks used for hanging IV fluid bags using a 3D printed fixture. The localizer is comprised of a speaker array with four speakers and a Bluetooth receiver (Fig. 2a). The localizer emits inaudible acoustic frequency modulated continuous wave (FMCW) signals that are captured by the microphone on the wearable device (Fig. 2b). The wearable device sends these captured acoustic signals via Bluetooth back to the localizer, which uses these signals to compute the 3D location of the wearable device with respect to the localizer.

To determine the hydrostatic reference point height, the distances between the wearable device microphone and three of the four localizer speakers are calculated. Triangulation is then used to compute the 3D location of the wearable device by extracting the time-of-flight information from the FMCW acoustic signals transmitted by the localizer. Computing accurate time-of-flight is challenging for two key reasons: (1) Time-of-flight computation requires that the two devices—the localizer and the wearable device—be continuously synchronized with each other to share a common clock. This is not trivial to achieve given the accuracy requirements for localization, considering sound only takes 0.03 ms to travel 1 cm. (2) Acoustic signals transmitted by the localizer reflect off nearby surfaces, objects and humans before arriving at the microphone. This is known as the multipath effect which makes it challenging to separate the direct line of sight path from all the reflections.

The above challenges are addressed using multiple technical components shown in Fig. 2c. First, to achieve accurate time synchronization between the two devices, a combination of Bluetooth-based (BLE) synchronization and a post-processing algorithm is used. Specifically, the localizer transmits time sync Bluetooth packets at a rate of 200 Hz with information about its free-running clock. When the wearable device receives this packet, it synchronizes its own clock. While this Bluetooth technique could substantially reduce the clock drift over time, it does not eliminate it because of sampling and precision limits. A postprocessing algorithm addresses this by performing linear regression on the height estimates over a longer duration to calculate the residual drift after Bluetooth synchronization. The received acoustic signal is then resampled using software to align with the timing of the transmitted signal at the localizer (see details in methods).

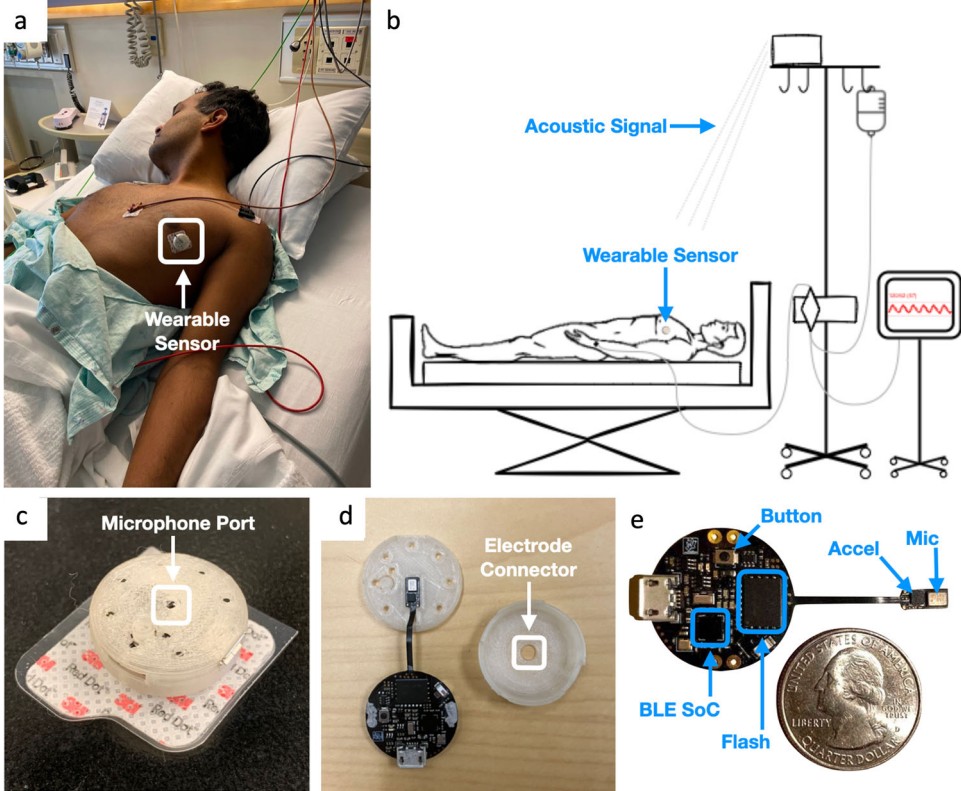

**Fig. 1 Height tracking system using a wearable acoustic device. a** Paper author demonstrates wearing the low-power wearable device for automated height tracking. Consent was obtained from the model for including this photograph in the manuscript. **b** System setup and diagram showing the wearable device capturing an acoustic signal from the localizer. Acoustic data is wirelessly transmitted back to the localizer for blood pressure correction. **c** Wearable device mounted on an electrocardiogram (ECG) electrode (3M Red Dot, Maplewood, MI); microphone is exposed via a port in the center of the wearable device. **d** Inside the wearable device, the device mounts to ECG electrode via the electrode connector. **e** Wearable device printed circuit board (PCB) with a Bluetooth Low Energy System-on-a-Chip (BLE SoC), flash integrated circuit (Flash), accelerometer (Accel), microphone (Mic), and a button for turning the device on.

To address multipath and achieve sub-centimeter time of flight computations, we built on our recent work[17] that introduces FMCW phase-based algorithms to disambiguate the direct path from multi-path reflections. However, this design[17] is focused on tracking a single speaker from a microphone array. The current design, in contrast, has the inverse setting; the wearable device has a single low-power microphone to reduce power consumption while the plugged-in localizer uses a more power consuming speaker array capable of calculating height measurements. The algorithms are adapted for this inverse problem to achieve accurate height computation for the wearable device. Finally, several techniques are utilized to minimize power consumption and address movement from people in the environment.

**Wearable device hardware design**. Our wearable device consisted of a pulse-density modulated (PDM) microphone (Invensense ICS-41350), a Bluetooth Low Energy (BLE) microcontroller (Nordic nRF52840), and an ultra low-power accelerometer (Bosch BMA400). The device was powered with a single CR2032 coin cell battery. The device's integrated PDM microphone was set to a clock frequency of 3.2 MHz. With an internal PDM decimation ratio of 64, this provided a sampling frequency of 50 kHz. Since near-ultrasonic frequencies were imperceptible to the adult human ear (18–22 kHz), a 50 kHz sampling rate was sufficiently high to avoid aliasing.

Two 16-bit 512 sample size Pulse-Code Modulation (PCM) buffers were round-robined: one was filled with incoming PCM data while the other was processed. The DMA was responsible for both clocking in the PDM data and converting it into PCM. One buffer was always connected to the DMA, while the other was freed for processing for the rest of the data pipeline[18]. When the buffer connected to the DMA was full, the buffers switched roles and we began processing data on the newly freed buffer, and connected the other buffer back to the DMA. With this design we always had a continuous PCM stream to operate on.

In order to maximize the device's battery life, we also incorporated an ultra low-power accelerometer. During periods of time when the device is static, we leveraged the accelerometer to turn off the Bluetooth and audio subsystems of the device. In this static state, we configured the BMA400 to operate in a low-power mode where it consumes as little as 160nA. In this operating mode, we configured the BMA400 for motion interrupts along the z-axis, so that the wearable device can enable the Bluetooth and microphone whenever the hospital bed or patient has been moved. In this manner, we duty cycle between our fully-on state (Bluetooth, accelerometer, microphone-enabled), and low-power state (accelerometer only). Power consumption of the wearable device components are shown in Table 1. The projected battery life in the fully-on state and low-power state is 36 h (6 mA) and 534 days (17.53 μA), respectively.

The wireless wearable device transmitted the PCM microphone and accelerometer data to the localizer device for computing the height information. To maximize throughput, however, we used the highest Bluetooth rate and packet sizes supported by Bluetooth 5.0, which is 2 Mbps and 247 bytes, respectively.

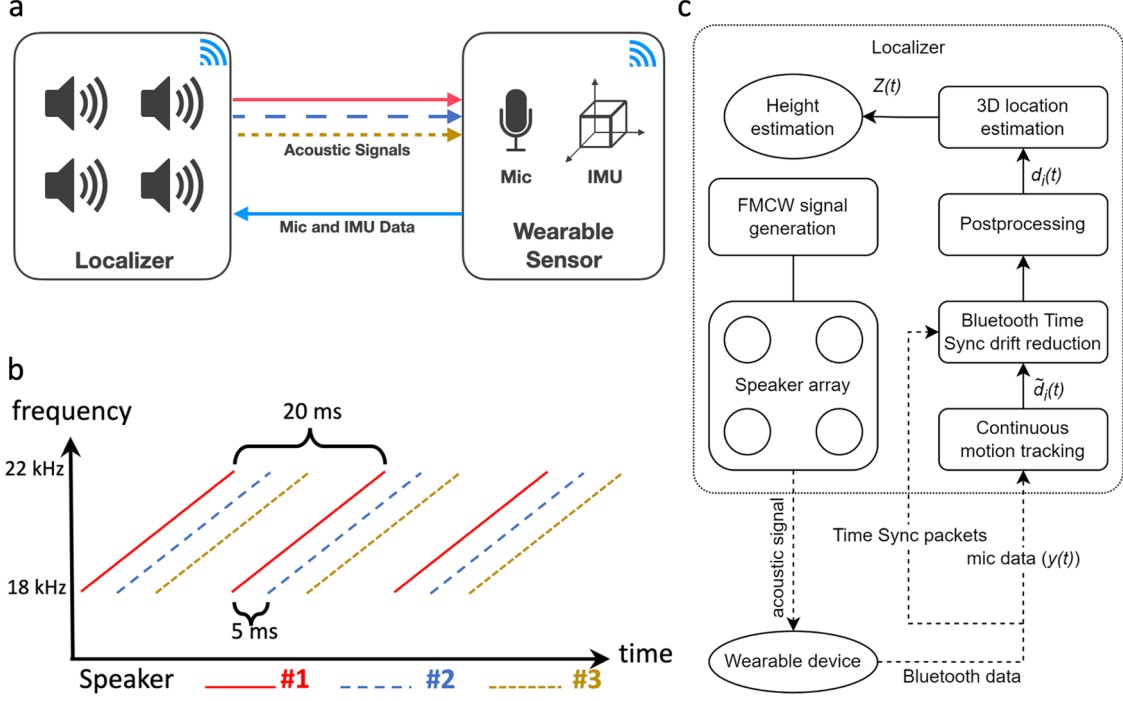

**Fig. 2 Algorithms and systems for tracking height. a** Localizer concurrently emits acoustic signals from three of its speakers, which are recorded by the wearable device. The wearable device then wirelessly transmits the recorded acoustic signals and Inertial Measurement Unit (IMU) data back to the localizer for processing. **b** Each of the speakers transmits a frequency modulated continuous wave (FMCW) signal. The signals from the three speakers are shifted in time by 5 ms. **c** Data pipeline with various algorithmic components for the localizer's height tracking system.

| Table 1 Wearable device power consumption | |
|---|---|
| **Component** | **Power consumption** |
| BLE SoC (nRF52840) | 12.02 mW |
| Microphone (ICS-41350) | 0.77 mW |
| Accelerometer (BMA400) | 6.3 µW |
| Ideal Diode (LM66100DCKT) | 0.27 µW |
| Buck Efficiency Loss (MAX38640) | 1.75 mW |
| Total | 14.55 mW |

The hardware schematic and layout for the wearable device were designed using the open source eCAD tool KiCad. A 2-layer flexible printed circuit was fabricated and assembled by PCBWay. The 3D printed enclosures were designed using AutoDesk Fusion 360 and printed with a Phrozen Sonic Mini using a liquid resin fabrication process. The MEMS microphone sits behind an exposed lid on the wearable device's outer surface. Finally, we designed a universal mounting port on the other side of the wearable device so that our system can be connected to the ECG electrodes that are commonly used in the clinical setting.

**Localizer hardware design**. The localizer transmitted FMCW chirps from its 4 speakers, geometrically separated onto the corners of a square. In addition, the localizer was responsible for establishing a wireless connection to the wearable patch, in order to receive the microphone data closest to the patient's chest. The four FMCW chirps were generated using a Teensy 3.6 microcontroller, and sent physically over a digital time-division multiplexed (TDM) audio bus to a Cirrus Logic CS42448 codec. While I2S is a standard stereo digital audio interface, our system necessitates 4 channels, so we opted for TDM which has support for up to 8 channels. Once the audio signals reach the codec, it outputs single-ended analog audio for each of our speakers. We

used an nRF52840 for time synchronization between the wearable device and the localizer, and a Raspberry Pi 4 for computation.

The localizer is synchronized with the wearable device using time sync beacons provided by the Nordic library[19]. The microphone data transmission and the time sync work independently. The time sync works by transmitting the timestamps of the localizer to the wearable device, and the wearable device can adjust its own clock to be synchronized with the localizer's clock. With a synchronized clock, we knew the exact timing of each recorded acoustic sample which was essential to achieve an accurate position estimation.

Finally, the entire system was mounted onto a 3D-printed fixture which is compatible with the standard IV poles, shown in Supplementary Fig. 1, so that our localizer can receive signals reliably from the wearable device. The top of our fixture (Supplementary Fig. 1) allowed the localizer to rest and be zip-tied down to the fixture for stability. The fixture and localizer can then be moved to any standard IV pole, where the bottom of our fixture mates to the top of an IV-pole. While the fixture was stable on the IV pole, we also provided velcro as an additional measure for the localizer to remain fixed onto the IV pole.

**Time synchronization algorithm**. A key requirement in our system was that the two devices—the localizer and the wearable device—need to be continuously synchronized with each other to share a common clock. This was not trivial to achieve given the accuracy requirements for localization, considering sound only takes 0.03 ms to travel 1 cm. We used a combination of Bluetooth-based synchronization and a post-processing algorithm to achieve accurate time synchronization between the two devices.

*Bluetooth-based time synchronization*. In our system, both the wearable device and localizer had a 32 MHz clock source with a ± 20 ppm frequency tolerance. So, in the worst case scenario, both devices had 2.4 milliseconds of drift per minute. To

maintain synchronization, we used the Nordic's TimeSlot API[19], which provided us access to the underlying radio hardware in between Bluetooth transmissions and gave us a transport to transmit and receive accurate time sync beacons[18]. Both our localizer and wearable device kept a free-running 16 MHz hardware timer with a maximum value of 800,000, overflowing and wrapping around at a rate of about 20 Hz. The localizer was assigned as the timing controller while the wearable device synchronized its free-running timer to the controller's timer. The localizer (timing controller) transmitted time sync packets at a rate of about 200 Hz. These packets contained the value of the free-running timer at the time of the radio packet transmission. When the wearable device received this packet, it then synchronized its own clock by adding or subtracting the offset to its own free-running timer. Once we had established a common clock between the localizer and the wearable device, the received acoustic signal were resampled to align with the timings of the transmitted signal at the localizer, and the drift was corrected. Specifically, we resampled the signal by performing an FFT followed by an inverse FFT using a frequency-shifted basis:

$$X_f = \sum_t x_t e^{-i2\pi ft/T} \tag{1}$$

$$\hat{x}_t = \sum_f X_f e^{i2\pi\alpha ft/T} \tag{2}$$

Here $\alpha$ is the drift factor ($\alpha = 1$ indicates no drift), $x_t$ is the original time domain signal and $\hat{x}_t$ is the resampled signal.

*Post-processing algorithm.* While the above Bluetooth time synchronization technique could substantially reduce the timing drift over time, it did not eliminate the timing drift because of sampling and precision limits. To further improve the accuracy, we designed a postprocessing algorithm that was applied to the signal after the Bluetooth time synchronization algorithm. We leveraged the observation that in target clinical settings, body motion was sparsely distributed in the time domain in short bursts. For example, within an operation that spans a few hours, the patient was only moved 10–20 times and each time the movement took a few seconds. Moreover, within any 5-min period, the body motion took much less than half of the duration.

To leverage this, we maintained a history of the last 5 min of the height estimation data. We split the 5-minute data into 30-s chunks, and calculated the residual square sum after a linear regression using each 30-s chunk. We sorted these residual values, chose the lower half of them, and calculated the mean slope of the corresponding linear regression results. The mean slope is the estimated drift factor that was then applied to cancel out future drifts.

**Height computation algorithm.** The key component of our automated height change tracking was an acoustic 3D localization system, where the relative position of the wearable device and the localizer attached to the pole was estimated in real time. The microphone equipped inside the wireless device received the acoustic signals emitted from the multiple speakers on the localizer. We only use three out of the four speakers to emit signals while leaving the remaining one as redundancy, if one of the three does not work properly. Given the geometry of the speaker array, any three subsets of them could be used to obtain the 3D location and more specifically the height difference between the two devices.

Specifically, the $i$-th speaker emits a time-shifted version of a frequency modulated chirp signal:

$$x_i(t) = \exp\left(j2\pi\left(f_0\left(t - \frac{iT}{5}\right) + \frac{f_1 - f_0}{2T}\left(t - \frac{iT}{5}\right)^2\right)\right) \tag{3}$$

where $x_i(t)$ is the emitted signal from speaker $i$ at time $t$ in the time domain, $T$ is the chirp duration, and $f_0$ and $f_1$ are the starting and ending frequency respectively. The chirps from the speakers were delayed by a multiple of $\frac{T}{5}$. In our implementation we set $f_0, f_1$ and $T$ to 18 kHz, 22 kHz and 43 ms respectively.

The received signal at the wearable device can now be written as the summation of the signals from all the speakers:

$$y(t) = \sum_{i=1}^{4} \sum_{p \in paths_i} \alpha_p x_i\left(t - \frac{dist(i,p)}{c}\right) + N(t) \tag{4}$$

Here $paths_i$ is the set of all the propagation paths from speaker $i$ on the localizer device to the microphone on the wearable device. $dist(i, p)$ is the distance from speaker $i$ to the microphone for path $p$, $\alpha_p$ is the attenuation factor of path $p$, $c$ is the speed of sound, and $N(t)$ is the random noise that includes the hardware noise as well as environmental noise.

The wireless device sent this signal, $y(t)$, back to the localizer device using Bluetooth. The localizer device used a two-stage process to estimate the relative change in the distance of the direct path from each speaker to the microphone. The arrangement of the speakers is fixed with each located at a corner of a rectangular panel, as shown in Fig. 2c.

During setup, we performed an initialization step where we measure the initial position of the wireless device related to the speaker array by placing the wireless device right next to the speaker array (localizer) at a specific known location. We then calculated the initial distance reported by the algorithm between each speaker and the wearable device, denoted as $D_i$, for the $i$th speaker. During its operation, the algorithm estimated the distance changes between the path and each of the three working speakers over time, denoted as $\widetilde{d}_i(t)$. We then obtained the three corresponding absolute distances after time synchronization and postprocessing, $d_i(t) = \mathbf{postprocess}(\mathbf{timesync}(\widetilde{d}_i(t) - D_i))$, and their averages, $\bar{d}(t) = \frac{\sum_i d_i(t)}{3}$.

The 3D location of the wearable device relative to the speaker array can then be estimated using the differences in the distance measurements as follows:

$$X(t) = \frac{\bar{d}(t)(d_1(t) - d_2(t))}{L} \tag{5}$$

$$Y(t) = \frac{\bar{d}(t)(d_1(t) - d_3(t))}{L} \tag{6}$$

$$Z(t) = \sqrt{\bar{d}(t)^2 - X(t)^2 - Y(t)^2} \tag{7}$$

where $L$ is the separation distance between adjacent speaker pairs, and $Z(t)$ is the height estimation.

Thus, if we can accurately compute the 1D distance, $\widetilde{d}_i$ over time, between the microphone and the three speakers, we can measure the 3D location of the wireless device. In the rest of this section, we describe how we computed the 1D distance between a microphone and a speaker. The key challenge in accurately computing the 1D location was that, in addition to the direct path from the speaker to microphone, the microphone also received reflections from all the objects and people in the environment (distant multipath) as well as reflections from the patients body close to the wireless path (body multipath). Thus, our algorithm should accurately identify the direct path in the presence of all these reflections. To do this we used a two step process.

*Step 1: Removing distant reflections with an band-pass filter.* Using the received signal $y(t)$ within each chirp cycle of duration $T$, we first decouple the four separate chirps that are emitted from each speaker respectively. This can be done by extracting the four peaks of the demodulated signal in the

frequency domain using a discrete Fourier transform similar to our prior FMCW processing algorithm[17]. For each chirp $x_i(t)$ from speaker $i$, we then applied a Finite Impulse Response (FIR) filter with the goal of leaving only a narrow range of frequency bands around the peak. To do this, we adaptively changed the delay of the FIR filter using the signal-to-noise ratio (SNR) of the received acoustic signals where when the SNR is more than 10 dB, we set the delay to 15 ms; at lower SNRs we used a larger delay of 30 ms. These parameters were not adjusted to favor a specific environment, and that fine-tuning them could further improve accuracy. Our evaluations are performed in actual hospital rooms. However, other locations with acoustically reverberant fixtures or loud interfering background sound may lead to worse results.

*Step 2: Removing the body multipath using FMCW phase.* The above process separates the received chirps from each speaker and reduces the impact of distant multipaths from objects and people in the environment that have a much different distance than the direct path. This left us with residual indirect paths from reflections from the human body, around the wireless device. When there were no consequential reflecting occlusions (not including clothes) between the wireless device and the localizer, the sum of the residual indirect paths had a lower amplitude than the direct path. Thus, we can extract the distance corresponding to the direct path using the FMCW phase on the output signal from Step 1. Specifically, the phase of the receiver FMCW signal can be written as[17],

$$\phi(t) \approx -2\pi \left( \frac{B}{T} t t_d + f_0 t_d - \frac{B}{2T} t_d^2 \right) \tag{8}$$

Here $t_d$ is the time of arrival of the direct path and $B$ is the bandwidth of the chirp signal, 4000 Hz. Hence, for each speaker $i$, by sampling the phase of the filtered signal corresponding to the speaker $i$ derived from the last step at a given time $t$ (e.g., $t = T/2$), we solved the above quadratic equation to extract potential solutions for $t_{d,i}$. We computed the $t_{d,i}$ for the direct path as the solution that is in the range of the FMCW chirp, $[0, T]$. We then computed the 1D distance (with an offset) between the speaker and microphone as $d_i = ct_{d,i}$, where $c$ is the speed of the acoustic signal in air.

**Clinical testing**. Written informed consent was obtained from 26 patients for this study which was approved by the Institutional Review Board at the University of Washington (STUDY00009667). English speaking patients who were undergoing cardiac surgery at the University of Washington and required invasive arterial blood pressure monitoring were eligible and consented prior to surgery. Up to two study sessions were performed for each patient, one with the endotracheal tube in place and mechanical ventilation and one without the endotracheal tube and with spontaneous ventilation. For the purpose of the study, we used the following two transducers:

*Clinical transducer.* This transducer was taped to the patient's skin at the level of the heart (hydrostatic reference point) with sufficient tubing length to move up and down with the patient during bed movements. While taping the transducer directly to the patient is not desirable, in our clinical study setup, we used this to provide ground truth data.

*Stationary transducer.* This transducer was attached to an IV pole next to the bed, which is the desired setup in these scenarios. It was initially set at the lowest bed height and did not move when the patient's bed moved.

We note that even when the two transducers were placed at the same height, they did not provide the same reading. For that reason, the stationary transducer measurements were normalized to the clinical transducer to account for baseline differences in transducer readings as follows:

$$\Delta transducer = MAP_{c,0} - MAP_{p,0} \tag{9}$$

Here $MAP_{c,0}$ is the mean measurement from the clinical transducer when attached to the patient's skin at the lowest bed height, and $MAP_{p,0}$ is the mean measurement from the stationary transducer.

Our algorithm adjusted the measured arterial pressure from the stationary transducer to account for the additional hydrostatic pressure due to the height difference between the patient's hydrostatic reference point and the transducer. This can be achieved by multiplying the height difference (in cm) by a factor of 0.735 to convert it to mmHg[20].

Specifically, we recorded three different mean arterial measurements as part of our clinical study.

*Clinical mean arterial pressure.* The clinical mean arterial pressure is the mean arterial measurement, $MAP_c$, that was used as a part of clinical care for the duration of the study and derived from the transducer taped to the patient so it moved as the bed moved up and down.

*Laser distance tool mean arterial pressure.* The laser distance tool mean arterial pressure is the calculated mean arterial pressure, $MAP_l$, using the stationary pressure transducer and the laser distance tool. Changes in distance were estimated based on changes in the distances output by the laser tool situated beneath the patient's bed. If $h_{l,x}$ represents the height measured by the laser in cm, $h_{l,0}$ represents the height measured by the laser distance tool at the lowest vertical setting on an ICU bed and $MAP_s$ is the pressure reading from the stationary transducer, then,

$$\Delta h_l = h_{l,x} - h_{l,0} \tag{10}$$

$$MAP_l = MAP_s - 0.735 \times \Delta h_l + \Delta transducer \tag{11}$$

*Automated height tracker mean arterial pressure.* The automated height tracker mean arterial pressure was the calculated mean arterial pressure, $MAP_r$. In this case, the distance between the hydrostatic reference point was estimated from the vertical measurements obtained from our height tracking system with the wearable device placed on the patient skin at a height of $h_{a,x}$ from the localizer device. $h_{a,0}$ represents the height of the wearable device at the lowest vertical setting on an ICU bed. Our estimated blood pressure was calculated as follows:

$$\Delta h_a = h_{a,x} - h_{a,0} \tag{12}$$

$$MAP_a = MAP_s - 0.735 \times \Delta h_a + \Delta transducer \tag{13}$$

**Statistics and reproducibility**. Twenty-six human participants were included in this study. The mean age of the patients was 56.8 (ranging from 24 to 80 years) with male:female ratio of 0.55. All patients were undergoing open cardiac surgery and thus had a major cardiac ailment. As the participants were in the intensive care unit, measurements were typically not replicated so as to minimize the impact on patient care. If the patient was in the intensive care unit the following day and the arterial line remained in place, a second data set was obtained when possible (3 cases). Data were evaluated using Excel and Python.

**Reporting summary**. Further information on research design is available in the Nature Portfolio Reporting Summary linked to this article.

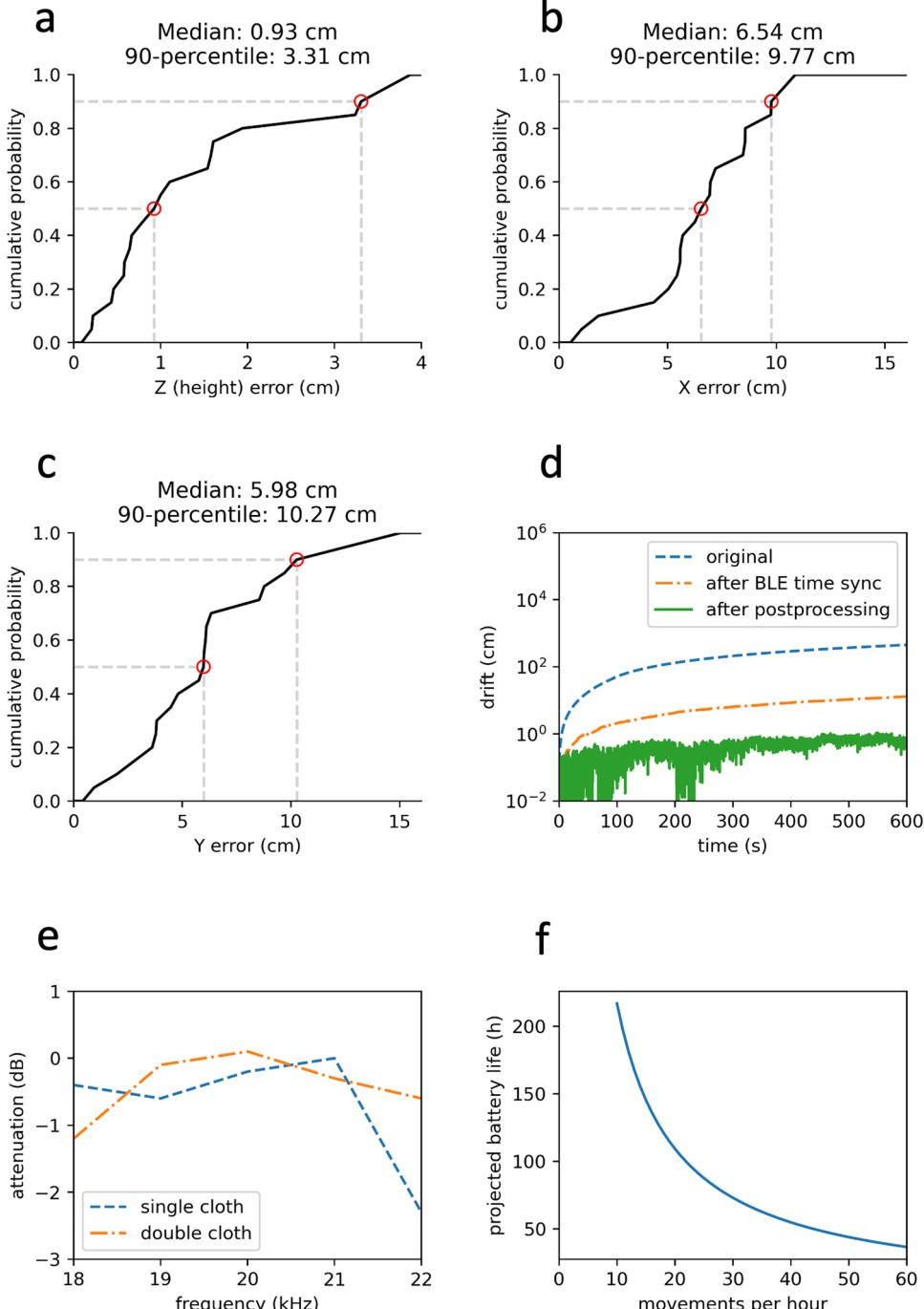

**Fig. 3 Benchmark results. a–c** The absolute localization error in the x, y and x axis respectively within a distance range from 0.2 m to 2 m. **d** The long-term accumulated drift before and after the two correction methods. **e** The attenuation of our acoustic signals in the target frequency band from one or two layers of cloth. **f** Battery life estimation with different movement frequencies.

## Results

**Benchmark testing**. The system was evaluated in a laboratory setting by placing the wireless device and the localizer at known positions and measuring the location estimation errors. The wireless device was placed at a distance of 0.2–2 m from the speaker array, and the absolute error of the location estimation was measured. Figure 3a–c shows the cumulative distribution function of the absolute error in the three dimensions. Because the speaker array is perpendicular to the height dimension, the height error is smaller than the other two dimensions. The median absolute height

estimation error was 0.93 cm which translates to less than 1 mmHg in anticipated blood pressure error.

Wireless synchronization between the wearable device and the localizer is important for accurate localization. Figure 3d shows the drift when the wearable device is stationary for 10 min. Two drift reduction algorithms were applied as described in the methods section. The plot shows that without any of the algorithms, the drift in the location estimation can be as high as 100 cm over 10 min. The plot also shows that drift is reduced after the Bluetooth time synchronization algorithm and even

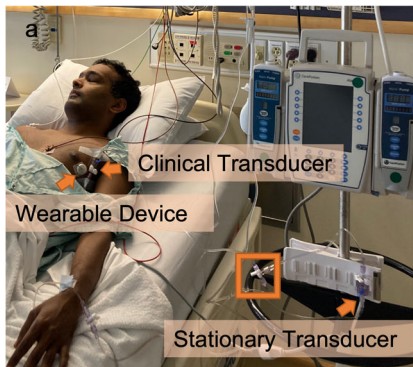
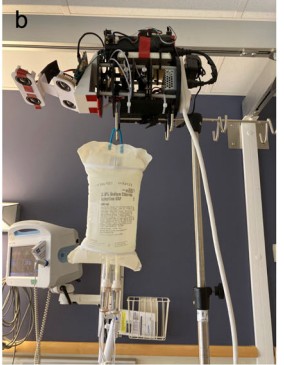
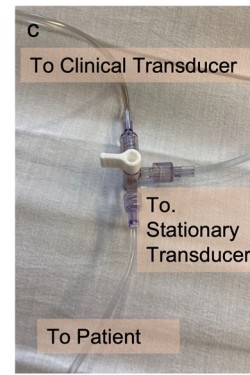

**Fig. 4 Two-transducer clinical evaluation setup. a** Paper author demonstrating patient positioning with the height tracking system in place for the clinical evaluation setup. Consent was obtained from the model for including this photograph in the manuscript. **b** The localizer included a 3D-printed dock for quick attachment to the top of an IV pole without impacting the ability to hang fluids from the pole. **c** A close-up visualization of the orange box shown in A that depicts the 3-way stopcock which allowed for simultaneous blood pressure measurements of the stationary and clinical pressure transducer from a single arterial catheter.

further reduced to the sub-centimeter level after applying the postprocessing method.

To validate the effect on the signal strength from clothing, Fig. 3e shows the attenuation from a single and double layer of medical grade cotton typically used in the operating room and the ICU. We can see that the attenuation is minimal and would not affect the operational range or accuracy. In addition, testing was performed in an operating room with multiple loud alarms going off from devices such as a ventilator, Bair hugger and pulse oximeter, to ensure they did not impact height measurements.

Finally, the battery life was estimated by measuring the current consumption with and without motion, and across different duty cycle settings. When the wearable device remains on under constant movement, the battery life is approximately 36 h. Battery life improves as a function of fewer movements per hour. While the localizer is powered by a wall power supply in this prototype, the wearable device is powered by a single coin cell battery. The wearable device has two primary operating modes: active and idle. In active mode, the accelerometer and microphone are both enabled, and the Bluetooth chip (nRF52840) is constantly streaming data, with consumption around 6 mA. However, power consumption can be reduced when the hospital bed or patient is stationary. In idle mode, when the accelerometer indicates that there is no motion, the wireless link and microphone are turned off, decreasing the wearable device consumption to as little as 17.53 µA. When the system needs to begin tracking movement, the accelerometer will provide an interrupt and wake the rest of the system to re-enter the active mode of operation.

**Clinical testing**. The system was evaluated using 243 height measurements obtained from 26 patients undergoing cardiac surgery who required invasive arterial pressure monitoring at the University of Washington Medical Center (UWMC). The study was approved by the Institutional Review Board and patients gave written, informed consent. Height and invasive arterial pressure measurements were performed in the ICU with the localizer affixed to the top of an IV pole which was placed adjacent to and above the patient's bed using a custom 3D printed mounting station as shown in (Fig. 4b). The wearable device was affixed to the patient's chest at a level approximating the right atrium of the heart (hydrostatic reference point) which was visually estimated as shown in (Fig. 4a)[21].

In order to establish the mean arterial pressure reference point with the bed at the lowest height, a pressure transducer was placed at the level of the patient's heart which was visually

estimated. This pressure transducer remained stationary during each patient's study session. Following a random pattern, the bed was raised and lowered to positions from 5 to 45 cm above the lowest height at 5 cm intervals as measured by the laser distance tool (Blaze GLM50C, Bosch, Gerlingen, Germany). This tool was placed on the floor under the patient's bed and directed up to measure the distance between the floor and the bottom of the bed. For each bed height, two mean arterial pressures were calculated. First, mean arterial pressure was calculated using the height difference between the lowest bed height and the new bed height as measured with the laser distance tool. A second mean arterial pressure was calculated using the height difference between the wearable device original position and the new position as measured by the height tracking system. Mean arterial pressure calculations involved converting the height difference into pressure difference and adding it to the stationary transducer mean arterial pressure measurement. Comparing the height differences measured by the wearable device and the laser distance tool was possible because the patient was supine during the study and if not sedated, directed not to move or shift in bed for approximately fifteen minutes of measurements. Thus, changes in bed height corresponded to height changes at the level of the wearable device. At each bed height, the mean arterial pressure was also measured by the clinical transducer taped to the patient's chest next to the wearable device which was the invasive arterial pressure used for clinical care.

Comparing the laser distance tool and the wearable device height differences resulted in a mean difference of $0.3 \pm 1.0$ cm with comparative results shown in Fig. 5a and cumulative probability of the difference shown in Fig. 5b. The mean difference in the mean arterial pressure calculated using the height difference measured by laser distance tool compared to the height difference measured by the height tracking system was $0.23 \pm 0.75$ mmHg; 90% of the measurements were within 1.2 mmHg (Fig. 5c, d).

*Two-transducer evaluation.* Two pressure transducers were connected in parallel as shown in Fig. 4a–c to allow for measuring invasive arterial pressures for clinical care and to maintain an arterial pressure reference point with respect to lowest bed height during study session. The non-compressible tubing from the pre-existing arterial catheter was connected to a 3-way stopcock at the end of the surgery. Using the 3-way stopcock, two sets of non-compressible tubing were connected to a pressure transducer used for clinical care and another pressure transducer used for arterial pressure reference point in the ICU, as shown in Fig. 4c. This experimental design with a 3-way stopcock opened to both transducers allowed for

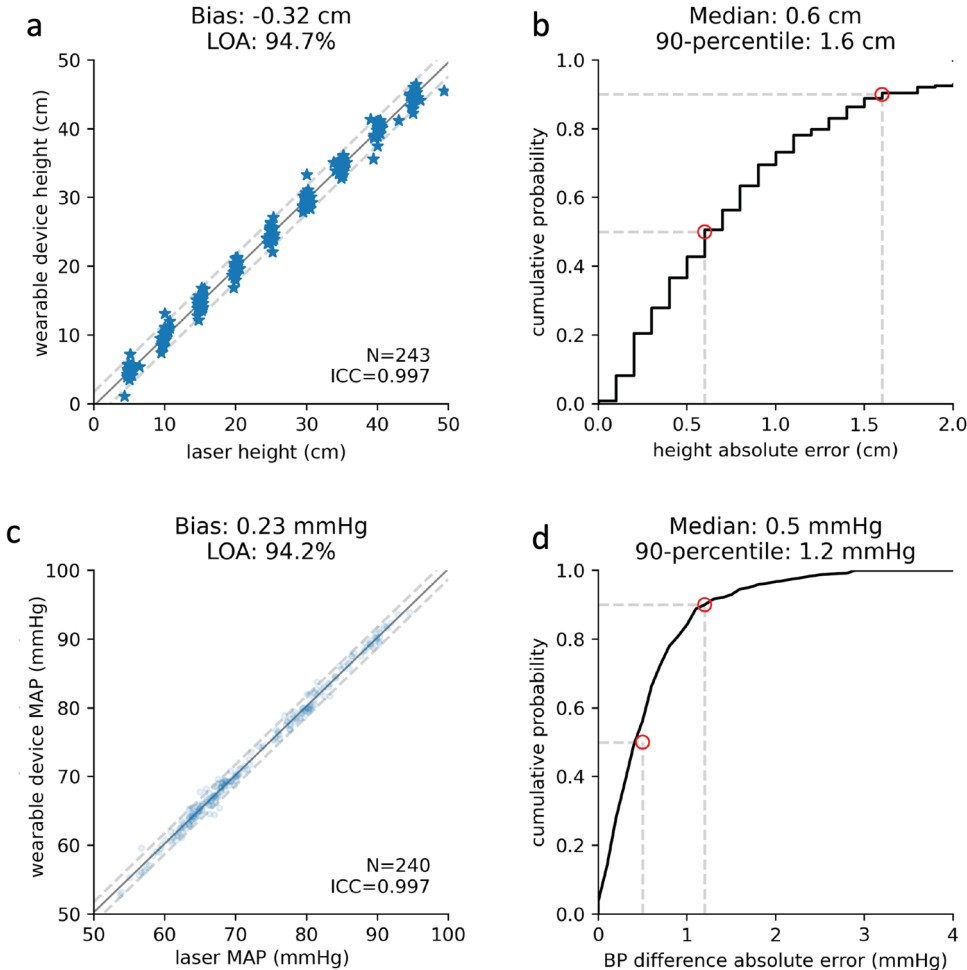

**Fig. 5 Clinical evaluation of height and pressure difference measurements. a** Comparison of measured height differences estimated by the automated height tracking system vs. laser distance tool. **b** Cumulative density function of the differences in height difference measurements from the height tracking system and laser distance tool. **c** Comparison of the estimated mean arterial pressure calculated using height differences measured by the laser distance tool vs. height tracker system. **d** Cumulative density function of the differences in mean arterial pressure for the height tracker system and laser distance tool. In **a**, **c**, the solid line shows the best linear fit while the dashed lines depict twice the standard deviation.

simultaneous invasive arterial pressure measurements from a single patient to be taken at two different heights at the same time. The pressure transducer used for clinical care was taped to the patient's chest at the level of the patient's hydrostatic reference point while the stationary transducer remained attached to the IV pole throughout all the measurements in the ICU.

At the start of each patient's study session, both transducers were placed at the same height on an IV pole, connected to separate monitors, and zeroed to atmospheric pressure. In addition to mean arterial pressure, systolic and diastolic blood pressures were recorded for each transducer to assess the intrinsic variation in the pressure transducers for 14 out of 29 (48%) of the study sessions. After the initial arterial pressure measurements with both transducers at the same height were recorded, the transducer used for clinical care was taped to the patient's chest at the level of the heart with the bed at its lowest height setting. The angular difference between the stationary and clinical transducers after the clinical transducer was taped to the chest was within 1.4 ± 1.2 degrees. An initial mean arterial pressure was measured for both transducers in this setting. At each height, mean arterial pressure was recorded for each of the transducers. At the conclusion, mean arterial pressure measurements were recorded continuously for two minutes to estimate a patient's baseline mean arterial pressure variability.

The mean arterial pressure measured by the clinical transducer, when compared to the calculated mean arterial pressure using the height tracking system, differed on average by 0.19 ± 2.8 mmHg as shown in Fig. 6a with 90% of the measurements within 4.3 mmHg (Fig. 6b). Given the use of a height conversion factor to calculate the pressure difference, it was important to assess for measurement bias as the distance from the lowest bed height increased. A linear fit of the difference between the clinical and calculated mean arterial pressure was performed and showed a slope of 0.4 mmHg over a height range of 50 cm.

Beat-to-beat mean arterial pressure variability is shown in Fig. 6c for each patient. One potential source of variation is physiologic beat-to-beat variation in blood pressure. All measurements were taken in quick succession but did not correspond to the exact same heartbeat, because the limited interface provided by the clinical device prevented it from achieving a beat-to-beat synchronization. Causes of beat-to-beat variation in blood pressure include intrathoracic pressure changes during respiration, variations in the RR interval (instantaneous variation in heart rate) and arrhythmias[22]. To account for the effect of beat-to-beat variation in blood pressure, we measured individual mean arterial blood pressure variation over the course of two minutes and separated the patients into high (greater than 4 mmHg) and low (less than or equal to 4 mmHg) variability groups. The

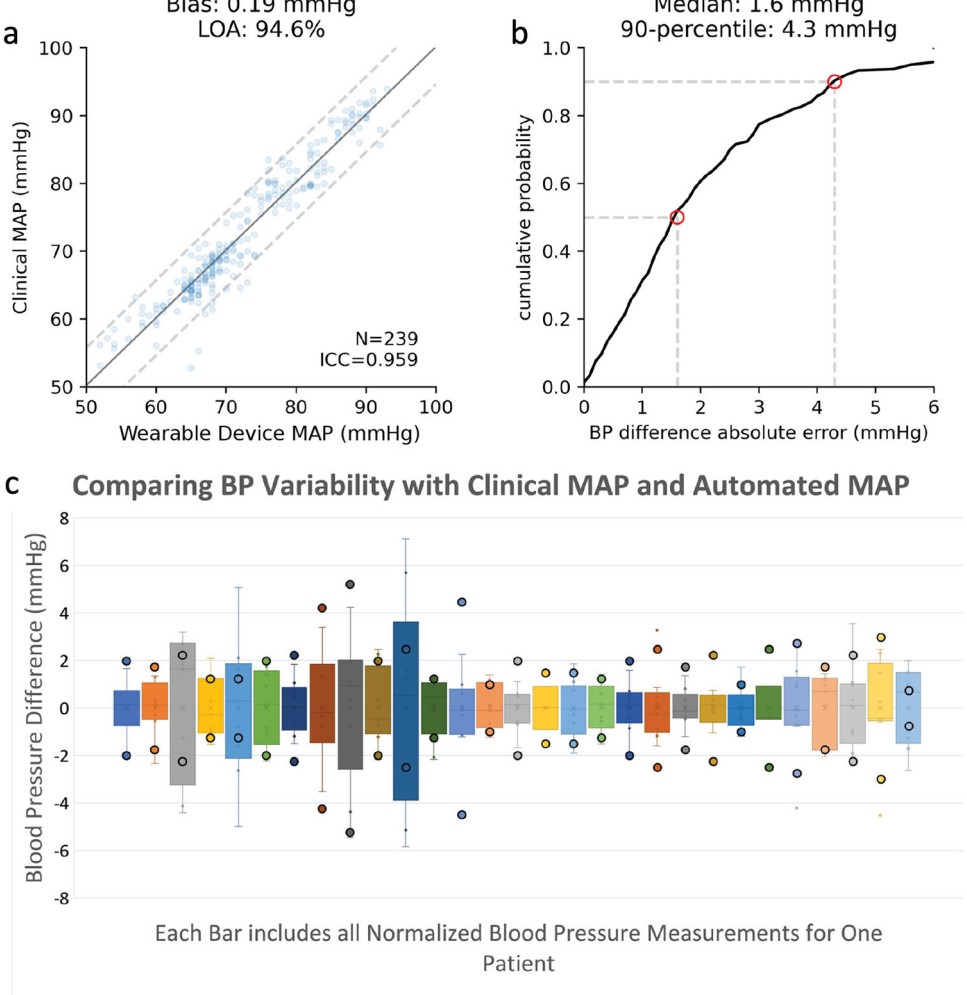

**Fig. 6 Results from two-transducer clinical testing: a** Comparison of the estimated mean arterial pressure calculated from the height tracking system vs. the clinical transducer mean arterial pressure. The solid line shows the best linear fit while the dashed lines depicts twice the standard deviation. **b** Cumulative density function of the differences in mean arterial pressure for the height tracking system and clinical measurement. **c** Comparison of mean arterial pressure variation over two minutes with the difference between the clinical and calculated mean arterial pressure for individual patient study sessions. Each box represents a different patient and includes the differences between the clinical transducer and the automated height tracking system for all measured heights (9), normalized to the average difference for all measurements for a given patient. Median values are shown as gray lines in between the 1st and 3rd quartile line. Whiskers extend to minimum and maximum values for each patient. The enclosed dots above and below each bar represent the range of mean arterial pressure values measured over a two-minute timespan using the clinical transducer. *MAP* mean arterial pressure.

standard deviation of the difference between the clinical and the calculated mean arterial pressure measurements using the height tracking system in the group with high beat-to-beat variability was 3.1 mmHg compared to 2.5 mmHg for the group with low beat-to-beat variability. The higher variability group demonstrated a larger standard deviation. Hence, it is likely that beat-by-beat mean arterial pressure variation contributed to the measurement differences we found between the height tracking system and clinical measurements which were not recorded during exactly the same beat.

Mean arterial pressure standard deviations were 3.0 mmHg for intubated patients and 2.2 mmHg for extubated patients. This may be due to the larger intrathoracic pressure changes during mechanical ventilation in comparison to spontaneous breathing[23].

## Discussion

In the clinical evaluation of our system, the median error in the height computation was 0.6 cm, corresponding to a pressure measurement difference well below 1 mmHg on average.

Furthermore, 95% of the measurements were within 2 mmHg when compared to the laser distance tool measurements. For mean arterial pressure, this is likely a clinically insignificant difference, indicating that this technology could be incorporated into invasive arterial pressure monitoring systems. It would provide automated arterial pressure corrections eliminating the need to manually change the transducer's height for patients requiring invasive arterial pressure monitoring. Our algorithms are capable of computing the BP adjustments at a rate of up to 60 Hz. However, update rates of even once per second, may not be clinically necessary in practice.

In this clinical study, the calculated mean arterial pressure measurements had a small mean difference of 0.19 mmHg and a larger standard deviation of 2.8 mmHg, This is within the International Organization for Standardization/American National Standards Institution (ISO/ANSI) standard with ± 3 mmHg accuracy for pressure transducers[10,24]. There are several factors that may contribute to the greater standard deviation for this metric. Each transducer used in our study has a known measurement variation of ± 1 mmHg per manufacturer[25]. Our

study setup requires two transducers to measure the ground truth. We often found this difference to be larger than 1 mmHg in many cases, (0.8 ± 1.9 mmHg), when the two transducers were positioned at the same height. This additional source of error could be due to the extra tubing length needed to connect the two transducers and allow them to be physically separated in space during measurements. There could also be differences in the amount of small air bubbles and blood clots within the tubing, tubing kinks or 3-way stopcocks angled slightly off from the expected position[26–29]. To minimize these errors, all lines were checked carefully, and only mean blood pressure values were reported rather than systolic and diastolic blood pressure. Representing the high and low points during the cardiac cycle, systolic and diastolic pressure can be impacted by variations in waveform damping (amplitude changes), while the mean value is less prone to these effects. In clinical use of this system, there would only be one transducer, so errors related to a secondary transducer would not cause discrepancies. Additionally, shorter tubing lengths could be used as the transducer would remain stationary which would reduce damping, resulting in a more optimal measurement.

Movement of the patient's chest related to breathing could also have led to some differences between the heights observed by the laser distance tool and our prototype wearable height tracking system. We measured the height of the patient's bed with a laser distance tool from the floor to the bottom of the bed frame while our wearable device was mounted to the anterior chest of the patient. Thus, the laser distance measurements do not account for chest movement during breathing. Studies on chest anterior to posterior dimension change with maximal inspiratory and respiratory effort for the average age of our study population is 3–3.5 cm[30]. Although patients have a smaller tidal volume and less chest wall motion after cardiac surgery than non-surgical participants during maximal respiratory effort, there are still changes in anterior to posterior chest dimensions which could cause small differences in height measurements. Thirty-second measurements of three stationary patients demonstrated height variation of 0.5 cm at a rate consistent with respiration, which corresponds to 0.4 mmHg of mean arterial pressure difference.

This study has the following limitations: For this systematic proof of concept analysis, early post-operative patients were studied, many sedated and mechanically ventilated, preventing extensive movement by the patient. While the benchmark testing demonstrates robustness with motion, future work will involve longer observation trials, including during physical therapy interventions to assess the reliability of measurements during voluntary patient movement as well as integration of automated blood pressure adjustments into the clinical vital sign monitor for real time readout. Other potential challenges include the movement of the pole that the localizer is attached to which could lead to inaccurate distance measurements. This base could be attached to the ceiling for more long-term studies. Further work should evaluate the potential of this technology for automating other invasive clinical pressure measurements such as central venous, pulmonary artery, cerebrospinal fluid or intracrainal pressure. Finally, while the device is designed to be compatible with ECG electrode that are regularly used in the clinical setting, further investigation is needed to determine the functionality of the device for long term use over multiple days.

While our acoustic localization algorithm assumes that there are no occlusions between the speaker array and the wearable device, we use the IMU data on the sensor to disambiguate between changes in the acoustic signals due to occlusions versus height changes of the wearable device itself. As a result, if the height of the wearable device has not changed but there is an occlusion of the direct path, the system does not update the height measurements since the IMU does not register any motion. When changes in the height of the wearable device occur (indicated by IMU data), the acoustic algorithm accurately updates the height information as long as there is no occlusion of the direct path for the duration when the height of the wearable device changes. If occlusions occur at the same time as changes to the height of the wearable device, our acoustic tracking system can run for an extended duration after the IMU triggers, until the occlusions clear up. This is done by continually computing the height until it stabilizes. Another way to mitigate this is to mount the localizer to the ceiling over the heart, minimizing the probability of occlusion events.

We anticipate the device to be most utilized in the ICU and operating room. Patients in the ICU are less likely to be sedated and may reposition themselves in the bed. They also potentially have visitors who can adjust the height of the bed without alerting a nurse. In addition, bed heights are frequently changed in the ICU as nurses perform various care tasks to minimize their own risk of occupational injury[31] and each height change requires the nurse to remember and adjust the pressure transducer for an arterial line that may be in place for days or even longer.

In summary, maintaining the transducer position for invasive arterial pressure measurements is a manual process that is prone to errors. In this study, we demonstrate a wireless, low-power system for automatically determining the hydrostatic reference point height with median precision of less than a centimeter. This prototype device could help decrease the task overload of healthcare providers and improve the accuracy of invasive arterial pressure measurements.

## Data availability

Source data for the figures are available as Supplementary Data 1. Additional data including systolic and diastolic pressure measurements can be obtained by contacting the corresponding author.

## Code availability

Code used to run the system can be found in the following repository: https://github.com/uw-x/bp-calibration[32].

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

## Acknowledgements

We acknowledge the work of Kishanee Haththotuwegama and Sharon Nguyen in obtaining patient consent, data collection, and figure editing. The project was conceived from an engineering innovation in health capstone project at the University of Washington. Funding is provided by the Laura Cheney Professorship in Anesthesia Patient Safety (A.B.), Foundation for Anesthesia Education and Research (K.M.), the Washington Research Foundation (K.M.), Washington State Society of Anesthesiologists (K.M.) and Moore Inventor Fellow award #10617 (S.G.).

## Author contributions

M.K, A.W., S.G. designed and built hardware; S.J., A.B., K.M. designed and performed the clinical testing; M.K., A.W. conducted the analysis with technical supervision by S.G., K.M.; M.K., A.W., S.G., K.M., A.B., S.J, interpreted the results; M.C, A.W., and K.M. generated the figures; S.G., K.M., A.W., and M.C. wrote the manuscript; S.G., K.M., M.K., S.J., A.B edited the manuscript.

## Competing interests

S.G. is a co-founder of Jeeva Wireless, Inc. and Wavely Diagnostics, Inc. The remaining authors declare no competing interests.
