## [Peer Review File · Communications Medicine]

Reviewers' comments:

Reviewer #1 (Remarks to the Author):

I found the manuscript very interesting and the study very well done. The manuscript is quite replete in detail regarding the engineering and data analysis which is very nice. My only fear is that clinicians without an engineering background may find the descriptions a bit overly detailed and technical. However, I suspect the audience is not front line clinicians just yet, until more advanced development is performed so I would not recommend any re-write for that audience.

I assume the range for the acoustic sensor and the microphones is not interfered with appreciably by other devices in the OR or ICU. Has this been verified that sound outputs from other devices do not overlap in this frequency range? This could contribute to some of the residual error found. You noted that anterior to posterior chest excursion could be a culprit given that the laser measurement was to the rigid bed frame while the sensor will always move rhythmically to some degree with respiration. Could you collect that rhythmic motion excursion and average it to further reduce standard deviation?

The one thing you did not describe in detail unless I missed it was the correction process for the display monitor. How frequently would the system "update" the BP on the monitor? I'm not sure it needs to be every second since clinicians aren't going to adjust BP management regimens that quickly.

I very much appreciate that the system was designed with the goal of being automatic instead of relying on an "alarm" to alert clinicians. Alarm fatigue is a huge problem and automation is key to reducing this problem and reserving alarms for where they are truly needed and can't be avoided through automated servo systems.

Finally, the Methods section in my document came after the Discussion. Is that typical for this journal? It is not the usual for medical journals. I assume the Editor would address that. Also, in the Results section there seemed to be a lot of stuff that seemed more appropriate for the Methods section. In fact, I would suspect that this manuscript may wind up with a longer Methods section than Results given that it is a descriptive paper of the development steps if everything that is Methodological was moved to the Methods section. However, I agree that the "story" may be better served in keeping some of the Methodological information in the Results section. I defer to the Editor on this part.

Otherwise, I found it an interesting method for addressing this common but often ignored problem and thankfully it avoids more alarms.

There are a few spelling errors and typos that need to be addressed through a proofread. Not many but a few.

Reviewer #2 (Remarks to the Author):

This paper developed a low-power wearable device that uses inaudible acoustic signals emitted from a speaker array to automatically compute height changes and correct the mean arterial blood pressure.

The strengths of the paper are 1) it automated the process of the measurement of the hydrostatic reference point and make corrections to the pressure measurements, and achieved higher accuracy than manual adjustment, and 2) the authors confirmed the

effectiveness of the method and device in a real clinical situation.

On the other hand, there are several questions to be clarified or described more:

1. What are the target accuracy for localization and pressure estimation? For example, in Page 11, the following sentences are written: "Comparing the laser distance tool and the wearable device height differences resulted in a mean difference of 0.3 ± 1.0 cm with comparative results shown in Fig. 4a and cumulative probability of the difference shown in 4b. The mean difference in the mean arterial pressure calculated using the height difference measured by laser distance tool compared to the height difference measured by the height tracking system was 0.23 ± 0.75 mmHg; 90% of the measurements were within 1.2 mmHg (Fig. 4c and d)." This gives the impression that the device achieved the target accuracies, but they are not explicitly provided. Indeed, in the section of discussion, some statements can be found, but they are still unclear to know the target accuracies.

2. The method estimates distance between the speakers and the microphone. However, it is not formulated how the distances D_i and $d_i(t)$ are estimated from the observed signal $y(t)$ around Page 23. This is important to know the property of the method as well as the validity.

3. Related to the above question, how about the robustness of the method against occlusion, e.g., when the direct path is hindered by the doctor or nurse to take care of the patient?

4. At Pages 21-22, a time synchronization technique is mentioned. I think that the authors refer to the following paper as another method to automate the calibration.

Katsutoshi Itoyama, Kazuhiro Nakadai: Synchronization of Microphones Based on Rank Minimization of Warped Spectrum for Asynchronous Distributed Recording. IROS 2020: 4842-4847

5. At Page 24, to deal with multi-paths, I found some magic numbers such as 10dB, 15ms, and 30ms, and the authors also mentioned that "the above process eliminated all distant multipaths from objects and people". This means that there are several assumptions on the environment. Please comment on if these are problematic when the environment changes.

6. At Page 23, do you assume any specific coordinates of microphone placements in Equations (1)-(3)?

7. What is "B" in Equation (4) at Page 24?

8. What is "MAP_s" at Page 26?

Reviewers' comments:

Reviewer #1 (Remarks to the Author): Critical Care, Clinical

I found the manuscript very interesting and the study very well done. the manuscript is quite replete in detail regarding the engineering and data analysis which is very nice. My only fear is that clinicians without an engineering background may find the descriptions a bit overly detailed and technical. However, I suspect the audience is not front line clinicians just yet, until more advanced development is performed so i would not recommend any re-write for that audience.

I assume the range for the acoustic sensor and the microphones is not interfered with appreciably by other devices in the OR or ICU.

Has this been verified that sound outputs from other devices do not overlap in this frequency range? This could contribute to some of the residual error found.

Thank you for this comment. We were also concerned about potential interference from other devices and some of our recordings in the submission were from operating rooms where we obtained recordings with the acoustic sensor while turning on as many alarms and sounds as we could (ventilator, Bair hugger, pulse ox etc). The presence of these additional sounds did not impact the recordings. This is as we expected since the acoustic device operates in the 17-22k Hz range, much higher than audible sounds present in the operating room. Further, the devices transmit a specific signal structure in a specific frequency which is unlikely to occur in other devices. We also note that in future versions of the device, we can increase or change the transmission frequencies to account for future devices that might interfere at these frequencies. A statement about these results has been added to the benchmark testing section of the manuscript.

You noted that anterior to posterior chest excursion could be a culprit given that the laser measurement was to the rigid bed frame while the sensor will always move rhythmically to some degree with respiration. Could you collect that rhythmic motion excursion and average it to further reduce standard deviation?

Thank you for this suggestion. After reviewing your comment, we went back to analyze the data in a more granular way and found some small changes in chest height over time as demonstrated by the graph below where we have taken 30 seconds of recordings at a given height and taken the average (T_{mean}), then plotted the difference between the height measured and the average at intervals of 0.1 seconds (T_x), demonstrating possible respirophasic changes. The total span of measured height difference is about 0.5cm, which corresponds to a change in mean arterial pressure of 0.4mmHg and could contribute to the standard deviation in the data. It is reassuring to note that respiratory changes are unlikely to have a clinically significant impact on mean arterial pressure recordings.

To address the above comment, the following was added to the manuscript: “Thirty second measurements of three stationary patients, demonstrated height variation of 0.5cm at a rate consistent with respiration, which corresponds to 0.4mmHg of mean arterial pressure difference.”

The one thing you did not describe in detail unless I missed it was the correction process for the display monitor. How frequently would the system "update" the BP on the monitor? I'm not sure it needs to be every second since clinicians aren't going to adjust BP management regimens that quickly.

In principle our algorithms can allow BP adjustments at a rate of 60 Hz. However, we agree that it may not be clinically necessary to adjust the BP values very quickly. The following statement was added to the manuscript: “Our algorithms are capable of computing the BP adjustments at a rate of up to 60 Hz. However, update rates of even once per second, may not be clinically necessary in practice.”

I very much appreciate that the system was designed with the goal of being automatic instead of relying on an "alarm" to alert clinicians. alarm fatigue is a huge problem and automation is key to reducing this problem and reserving alarms for where they are truly needed and can't be avoided through automated servo systems.

We agree, thank you.

Finally, the Methods section in my document came after the Discussion. Is that typical for this journal? It is not the usual for medical journals. I assume the Editor would address that. Also, in the Results section there seemed to be a lot of stuff that seemed more appropriate for the Methods section. In fact, I would suspect that this manuscript may wind up with a longer Methods section than Results given that it is a descriptive paper of the development steps if everything that is Methodological was moved to the Methods section. However, I agree that the "story" may be better served in keeping some of the Methodological information in the Results section. I defer to the Editor on this part.

Thank you for this feedback, we have changed the ordering to make sure that the methods come before the discussion.

Otherwise, I found it an interesting method for addressing this common but often ignored problem and thankfully it avoids more alarms.

There are a few spelling errors and typos that need to be addressed through a proofread. Not many but a few.

Thank you for this feedback, the manuscript has been thoroughly reread and typos addressed.

Reviewer #2 (Remarks to the Author): Acoustic signal processing, sound source localization

This paper developed a low-power wearable device that uses inaudible acoustic signals emitted from a speaker array to automatically compute height changes and correct the mean arterial blood pressure.

The strengths of the paper are 1) it automated the process of the measurement of the hydrostatic reference point and make corrections to the pressure measurements, and achieved higher accuracy than manual adjustment, and 2) the authors confirmed the effectiveness of the method and device in a real clinical situation.

On the other hand, there are several questions to be clarified or described more:

1. What are the target accuracy for localization and pressure estimation? For example, in Page 11, the following sentences are written: "Comparing the laser distance tool and the wearable device height differences resulted in a mean difference of 0.3 ± 1.0 cm with comparative results shown in Fig. 4a and cumulative probability of the difference shown in 4b. The mean difference in the mean arterial pressure calculated using the height difference measured by laser distance tool compared to the height difference measured by the height tracking system was 0.23 ± 0.75 mmHg; 90% of the measurements were within 1.2 mmHg (Fig. 4c and d)." This gives the impression that the device achieved the target accuracies, but they are not explicitly provided. Indeed, in the section of discussion, some statements can be found, but they are still unclear to know the target accuracies.

The aim from our work was to obtain accuracy levels consistent with ISO/ANSI standards for blood pressure transducers, which is ± 3 mmHg. This was achieved as mentioned in the discussion. An additional statement for clarification was added to the introduction (second last sentence): "The goal of this study was to ensure that the height tracking system could accurately measure mean arterial pressure within 3mm Hg of the current clinical standard measurement while automatically adjusting for changes in heart height."

2. The method estimates distance between the speakers and the microphone. However, it is not formulated how the distances D_i and $d_i(t)$ are estimated from the observed signal $y(t)$ around Page 23. This is important to know the property of the method as well as the validity.

Thank you for the comment.

During setup, we performed an initialization step where we measure the initial position of the wireless device related to the speaker array by placing the wireless device right next to the speaker array (localizer) at a specific known location. We then calculated the initial distance between each speaker and the wearable device, denoted as D_i , for the i th speaker. During its operation, the algorithm estimated the distance changes between the path and each of the three working speakers over time, denoted as $\tilde{d}_i(t)$. We then obtained the three corresponding absolute distances, $d_i(t) = \tilde{d}_i(t) + D_i$, and their averages, $\bar{d}(t) = \frac{\sum_i d_i(t)}{3}$.

The 3D location of the wearable device relative to the speaker array can then be estimated using the differences of the distance measurements as follows:

$$X(t) = \frac{\bar{d}(t)(d_1(t) - d_2(t))}{L} \quad (1)$$

$$Y(t) = \frac{\bar{d}(t)(d_1(t) - d_3(t))}{L} \quad (2)$$

$$Z(t) = \sqrt{\bar{d}(t)^2 - X(t)^2 - Y(t)^2} \quad (3)$$

where L is the separation distance between adjacent speaker pairs, and $Z(t)$ is the height estimation.

In the above text from the paper, we explain how we compute D_i and $d_i(t)$.

3. Related to the above question, how about the robustness of the method against occlusion, e.g., when the direct path is hindered by the doctor or nurse to take care of the patient?

Thank you for this comment. This is indeed an important consideration. We have added the following text to the discussion to clarify this: "While our acoustic localization algorithm assumes that there are no occlusions between the speaker array and the wearable device, we use the IMU data on the sensor to disambiguate between changes in the acoustic signals due to occlusions versus height changes of the wearable device itself. As a result, if the height of the wearable device has not changed but there is an occlusion of the direct path, the system does not update the height measurements since the IMU does not register any motion. When changes in the height of the wearable device occur (indicated by IMU data), the acoustic algorithm accurately updates the height information as long as there is no occlusion of the direct path for the duration when the height of the wearable device changes. If occlusions occur at the same time as changes to the height of the wearable device, our acoustic tracking system can run for an extended duration after the IMU triggers, until the occlusions clear up. This is done by continually computing the height until it stabilizes. Another way to mitigate this is to mount the localizer to the ceiling over the heart, minimizing the probability of occlusion events."

4. At Pages 21-22, a time synchronization technique is mentioned. I think that the authors refer to the following paper as another method to automate the calibration.

Katsutoshi Itoyama, Kazuhiro Nakadai: Synchronization of Microphones Based on Rank Minimization of Warped Spectrum for Asynchronous Distributed Recording. IROS 2020: 4842-4847

Thank you for pointing this out. We did not use the above mentioned method. The paper mentioned by the reviewer is a valuable reference of a general solution to the asynchronous microphone synchronization problem. In our scenario, because we know the Bluetooth timestamp and the sparsity of the movement, we use a simpler method instead as we described in our paper under the time synchronization subheading, and refer to our previous work, citation [11].

5. At Page 24, to deal with multi-paths, I found some magic numbers such as 10dB, 15ms, and 30ms, and the authors also mentioned that "the above process eliminated all distant multipaths from objects and people". This means that there are several assumptions on the environment. Please comment on if these are problematic when the environment changes.

Thanks for pointing this out. We have now provided some clarification in the manuscript. We have also edited this sentence to be more precise: the above process reduces the effect of distant multipaths from objects and people since they arrive at a much longer time delay than the direct path.

These parameters were set while designing the algorithm in a lab setting which was different from the ICU or operating rooms used in evaluation. As a result they have not been adjusted to favor a specific environment, and in principle, could be further fine-tuned to improve the accuracy. Our evaluations are performed in real ICU rooms which are representative to other ICU and operating rooms in a hospital. To ensure that readers do not take our claims to imply that it will work in all possible acoustic environments, we acknowledge that rooms with acoustically reverberant fixtures or loud interfering background sound may lead to worse results and adapting those parameters may improve performance.

6. At Page 23, do you assume any specific coordinates of microphone placements in Equations (1)-(3)?

We assume that the speakers that are part of the array are placed in the corners of a rectangular panel, as shown in Fig. 2 (C). We have added this detail to the description of the equations.

7. What is "B" in Equation (4) at Page 24?

B is the bandwidth of the chirp signal, which is 4000 (Hz). Thanks for pointing this out and we have added this to the description of the equation.

8. What is "MAP_s" at Page 26?

MAP_s is the mean arterial pressure measured by the stationary pressure transducer and is now defined in the paper.

Reviewers' comments:

Reviewer #1 (Remarks to the Author):

My questions and suggestions have been adequately addressed. the manuscript is ready for editor review.

Reviewer #2 (Remarks to the Author):

Thank you for the clarification. Almost all comments from me seem to be clarified except for one. You mentioned that the original text includes the answer for my second comment. I assume that $y(t)$ is an input to the system and $d_i(t)$ will be estimated from $y(t)$ if I correctly understand the basic procedure of the proposed method. However, no formulation on how to use $y(t)$ in the system is disclosed. I strongly recommend that 1) clarification of this point in the text, and 2) adding these variables including $y(t)$ to Fig. 2(c). For example, acoustic signal from speaker array can be denoted as $x_i(t)$, and input to wearable device and mic data can be denoted as $y(t)$. So, it is preferable and understandable when such variables are explicitly written in the figure. The input and output of each module in the process flow like Fig. 2(c) is very important and helpful to understand the proposed method.

There are additional suggestions as follows:

* L.1 P.12: i th \rightarrow i -th

* L.4 P.12: f_0, f_1 are \rightarrow f_0 and f_1 are

* L.9 P.12: when you use LaTeX, \left and \right are missing for brackets for $(t\text{-dist}(i,p)/c)$

* L.17 P.12: Fig.2C \rightarrow Fig.2(C)

* Eq.(4): \left and \right are missing for brackets

* Notation styles for subfigures in Figs. 2 and 3 are inconsistent. (A), (B) located in the bottom of each subfigure in Fig.2 vs A, B located in the top-left corner in Fig. 3.

Usually, captions should be located in the bottom of each subfigure. It will be simple and easy to maintain such a consistency when you use subfigure environment in LaTeX (see [https://ja.overleaf.com/learn/latex/How_to_Write_a_Thesis_in_LaTeX_\(Part_3\)%3A_Figures%2C_Subfigures_and_Tables#Subfigures](https://ja.overleaf.com/learn/latex/How_to_Write_a_Thesis_in_LaTeX_(Part_3)%3A_Figures%2C_Subfigures_and_Tables#Subfigures))

* P.19: "*" is used as a multiplication operator. In signal processing, it tends to be used as convolution, so \times or \cdot would be better.

Reviewers' comments:

Reviewer #1 (Remarks to the Author):

My questions and suggestions have been adequately addressed. the manuscript is ready for editor review.

Thank you for your review and comments.

Reviewer #2 (Remarks to the Author):

Thank you for the clarification.

Almost all comments from me seem to be clarified except for one.

Thank you for this comment.

You mentioned that the original text includes the answer for my second comment. I assume that $y(t)$ is an input to the system and $d_i(t)$ will be estimated from $y(t)$ if I correctly understand the basic procedure of the proposed method. However, no formulation on how to use $y(t)$ in the system is disclosed. I strongly recommend that 1) clarification of this point in the text, and 2) adding these variables including $y(t)$ to Fig. 2(c). For example, acoustic signal from speaker array can be denoted as $x_i(t)$, and input to wearable device and mic data can be denoted as $y(t)$. So, it is preferable and understandable when such variables are explicitly written in the figure. The input and output of each module in the process flow like Fig. 2(c) is very important and helpful to understand the proposed method.

Thank you for pointing this out. We have made changes to both the text and the figure to clarify how $d_i(t)$ is determined. The process is to first separate the four chirps in $y(t)$ from the four speakers respectively, and then filter out the distant multipaths. Afterwards, we use the phase information to derive the time-of-arrivals, which are used to calculate the distances. We have revised the text in the Height Computation Algorithm section to highlight the use of $y(t)$ and the two-step approach to obtain \tilde{d}_i as follows:

Step 1: Removing distant reflections with an band-pass filter. Using the received signal $y(t)$ within each chirp cycle of duration T , we first decouple the four separate chirps that are emitted from each speaker respectively. This can be done by extracting the four peaks of the demodulated signal in the frequency domain using a discrete Fourier transform similar to our prior FMCW processing algorithm [17]. For each chirp $x_i(t)$ from speaker i , we then applied a Finite Impulse Response (FIR) filter with the goal of leaving only a narrow range of frequency bands around the peak. To do this, we adaptively changed the delay of the FIR filter using the signal-to-noise ratio (SNR) of the received acoustic signals where when the SNR is more than 10 dB, we set the delay to 15 ms; at lower SNRs

we used a larger delay of 30 ms. These parameters were not adjusted to favor a specific environment, and that fine-tuning them could further improve accuracy. Our evaluations are performed in actual hospital rooms. However, other locations with acoustically reverberant fixtures or loud interfering background sound may lead to worse results.

Step 2. Removing the body multipath using FMCW phase. The above process **separates the received chirps from each speaker and** reduces the impact of distant multipaths from objects and people in the environment that have a much different distance than the direct path. This left us with residual indirect paths from reflections from the human body, around the wireless device. When there was no significant reflecting occlusions (not including clothes) between the wireless device and the localizer, the sum of the residual indirect paths had a lower amplitude than the direct path. Thus, we can extract the distance corresponding to the direct path using the FMCW phase on the output signal from Step 1. Specifically, the phase of the receiver FMCW signal can be written as [17],

$$\phi(t) \approx -2\pi \left(\frac{B}{T}tt_d + f_0t_d - \frac{B}{2T}t_d^2 \right) \quad (4)$$

Here t_d is the time of arrival of the direct path and B is the bandwidth of the chirp signal, 4000 Hz. **Hence, for each speaker i , by sampling the phase of the filtered signal corresponding to the speaker i derived from the last step at a given time t (e.g., $t = T/2$), we solved the above quadratic equation to extract potential solutions for $t_{d,i}$. We computed the $t_{d,i}$ for the direct path as the solution that is in the range of the FMCW chirp, $[0, T]$. We then computed the 1D distance (with an offset) between the speaker and microphone as $\tilde{d}_i = ct_{d,i}$, where c is the speed of the acoustic signal in air.**

In addition, Figure 2C has been updated to include the input and output of each module in the process flow.

There are additional suggestions as follows:

- * L.1 P.12: ith -> i-th
- * L.4 P.12: f_0, f_1 are -> f_0 and f_1 are
- * L.9 P.12: when you use LaTeX, \left and \right are missing for brackets for $(t\text{-dist}(i,p)/c)$
- * Eq.(4): \left and \right are missing for brackets

All of the above LaTeX changes have been made, thank you for your suggestions.

- * L.17 P.12: Fig.2C -> Fig.2(C)

* Notation styles for subfigures in Figs. 2 and 3 are inconsistent. (A), (B) located in the bottom of each subfigure in Fig.2 vs A, B located in the top-left corner in Fig. 3.

Usually, captions should be located in the bottom of each subfigure. It will be simple and easy to maintain such a consistency when you use subfigure environment in LaTeX (see

[https://ja.overleaf.com/learn/latex/How_to_Write_a_Thesis_in_LaTeX_\(Part_3\)%3A_Figures%2C_Subfigures_and_Tables#Subfigures](https://ja.overleaf.com/learn/latex/How_to_Write_a_Thesis_in_LaTeX_(Part_3)%3A_Figures%2C_Subfigures_and_Tables#Subfigures))

We have changed figure styles so that capital letters appear in the top-right corner of each figure. We changed from (A), (B) etc to A, B so to keep with this uniform styling, all figure descriptions have also been changed to capital letters without parenthesis.

* P.19: "*" is used as a multiplication operator. In signal processing, it tends to be used as convolution, so \times or \cdot would be better.

This is now changed to \times , thank you for the suggestion.

REVIEWERS' COMMENTS:

Reviewer #2 (Remarks to the Author):

All my comments are clarified now.